# Exploring the State of Machine Learning and Deep Learning in Medicine: A Survey of the Italian Research Community

**Alessio Bottrighi** [1,2,*,†] and **Marzio Pennisi** [1,2,*,†]

1 Dipartimento di Scienze e Innovazione Tecnologica (DiSIT), Computer Science Institute, Università del Piemonte Orientale, 15121 Alessandria, Italy

2 Laboratorio Integrato di Intelligenza Artificiale e Informatica in Medicina, Azienda Ospedaliera SS. Antonio e Biagio e Cesare Arrigo, Alessandria—e DiSIT—Università del Piemonte Orientale, 15121 Alessandria, Italy

* Correspondence: alessio.bottrighi@uniupo.it (A.B.); marzio.pennisi@uniupo.it (M.P.);
Tel.: +39-0131-360338 (A.B.); +39-0131-360186 (M.P.)

† These authors contributed equally to this work.

**Abstract:** Artificial intelligence (AI) is becoming increasingly important, especially in the medical field. While AI has been used in medicine for some time, its growth in the last decade is remarkable. Specifically, machine learning (ML) and deep learning (DL) techniques in medicine have been increasingly adopted due to the growing abundance of health-related data, the improved suitability of such techniques for managing large datasets, and more computational power. ML and DL methodologies are fostering the development of new "intelligent" tools and expert systems to process data, to automatize human–machine interactions, and to deliver advanced predictive systems that are changing every aspect of the scientific research, industry, and society. The Italian scientific community was instrumental in advancing this research area. This article aims to conduct a comprehensive investigation of the ML and DL methodologies and applications used in medicine by the Italian research community in the last five years. To this end, we selected all the papers published in the last five years with at least one of the authors affiliated to an Italian institution that in the title, in the abstract, or in the keywords present the terms "machine learning" or "deep learning" and reference a medical area. We focused our research on journal papers under the hypothesis that Italian researchers prefer to present novel but well-established research in scientific journals. We then analyzed the selected papers considering different dimensions, including the medical topic, the type of data, the pre-processing methods, the learning methods, and the evaluation methods. As a final outcome, a comprehensive overview of the Italian research landscape is given, highlighting how the community has increasingly worked on a very heterogeneous range of medical problems.

**Keywords:** artificial intelligence; machine learning; medicine; deep learning

## 1. Introduction

Currently, artificial intelligence (AI) is playing an increasingly important role in the medical field, which has ever represented in the past a source of challenges and an important area for both experimenting and developing AI methodologies. One of the first and most prominent AI research areas is machine learning (ML) [1]. Similar to medicine, for ML, the observation and analysis of data is fundamental. In the past century, the development and use of ML methodologies in medicine were, however, very limited, as presented in Figure 1. There are several reasons for such a limitation, including the fact that the application of AI in medicine was initially focused towards different approaches than ML, such as expert systems (e.g., [2]), and that the need for "large" amounts of data to automatically discover hidden patterns was unthinkable at the time.

**Figure 1.** No. of papers indexed by SCOPUS per year on machine/deep learning for the medical field (see Section 2.1 for details about the query).

In recent years, the advent of novel paradigms such as big data and the Internet of Things (IoT), capable of bringing huge amounts of novel heterogeneous data to feed ML/DL algorithms, as well as new computational models and the increased computational resources necessary to train complex algorithms in a reasonable time, allowed AI and, in particular, machine learning to become a growing phenomenon, both in the industrial and research fields. The adoption of AI technologies has given new opportunities but also new challenges and new issues. In particular, the advancement that AI can bring to the medical field can be related to:

- Diagnosis and Prognosis: the analysis of medical data can assist physicians in diagnosing diseases and predicting patient outcomes. Thus, earlier detection and personalized treatment plans could be possible;
- Drug Discovery: this could be accelerated by analyzing large amounts of biological and chemical data. Thus, new drug candidates and new potential treatments for various diseases could be identified;
- Personalized Medicine: treatment plans for individual patients can be tailored by analyzing genetic, molecular, and clinical data. Thus, more precise and effective treatments can be designed;
- Remote Monitoring and Telemedicine: AI-powered devices and applications can monitor patients remotely, enabling timely interventions and reducing the need (and the costs) for frequent in-person visits. Chronic disease management and rural healthcare have a particular benefit;
- Efficiency and Cost Reduction: tasks such as medical coding, billing, and administrative processes can be automated. Furthermore, the processes in a healthcare institution can be analyzed to find possible issues and/or be confronted with gold-standard(s). In this way, it is possible to increase efficiency and reduce costs;
- Patient Engagement: specific applications can provide patients with personalized health information, reminders, and recommendations, promoting proactive healthcare management;
- Clinical Trials: clinical trials can be analyzed, and potential patterns can be identified in large datasets, leading to insights that might not be evident through traditional analysis methods. Thus, the peace of clinical trials and of medical research can be accelerated.

However, new issues and challenges also arise:

- Integration: integrating AI systems into existing healthcare workflows and electronic health record systems may require significant changes to the infrastructure and processes.
- Lack of Transparency: Some AI models (such as DL models but also some ML models) are very complex and called black boxes due to their inability to provide (clear) explanations for their outputs (i.e., decisions). The lack of transparency can be a relevant barrier to gaining trust and acceptance from medical staff and from patients.
- Bias and Fairness: AI algorithms can inherit biases present in the acquired data used to train them. Thus, their output (i.e., medical decisions) could be affected by biases.
- Data Privacy and Security Concerns: AI in medicine relies heavily on patient data, which raises concerns about privacy and security;
- Regulatory Hurdles: the deployment of AI in medical practice is subject to rigorous regulatory processes to ensure patient safety and efficacy. Moreover, the legal regulation of the utilization of AI is in constant evolution. Navigating these regulatory and legal frameworks can be really time-consuming.

So, while new issues related to ethics, privacy, bias, and regulatory hurdles are posed, the application of ML and DL to the medical domain can lead to a consistent number of gains that cannot be overlooked.

By analyzing the scientific literature, we can claim that the maturity of AI methodologies has led to numerous results. In particular, looking at the production of scientific articles (see Figure 1), it is possible to see that the adoption of ML/DL methodologies has grown exponentially in the two last decades. This trend is particularly evident in the last 5 years (i.e., since 2018), and the Italian research community is one of the main worldwide players in this field, being the seventh country for the number of scientific papers indexed on SCOPUS (see Figure 2).

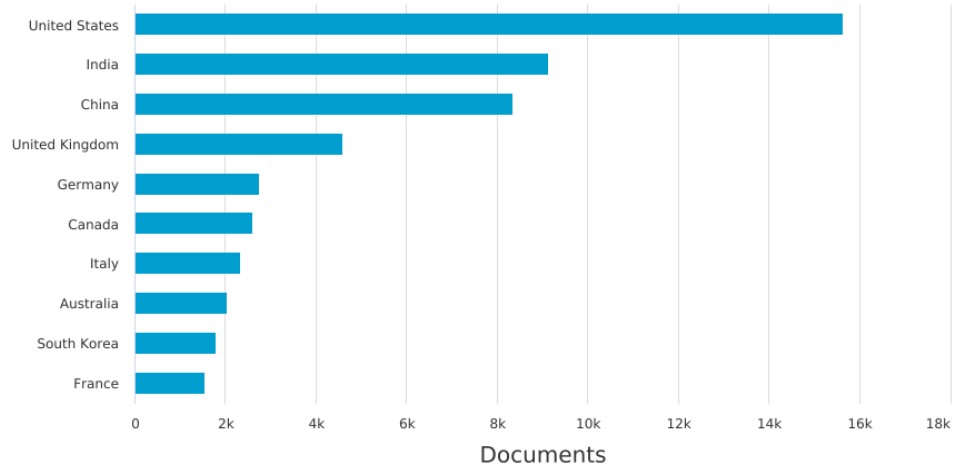

**Figure 2.** Papers by country on machine learning/deep learning for the medical field, indexed by SCOPUS.

To obtain a picture of the Italian scientific research in ML/DL in medicine, we carried out a systematic survey of the state of the art in Italy, according to the latest trends depicted in the scientific papers produced by the Italian community. In particular, we focused on the period starting in 2018. Notably, the query was performed on 13 January 2023, and we decided to consider also the papers that were published or in publication in 2023.

In summary, with the aim of taking a snapshot of the current Italian research community on ML, in this paper:

- We review the state of the art in Italy in recent years (i.e., since 2018), focusing on ML/DL in medicine, including all medical areas.
- We present a general map of the ML/DL research in Italy

- We propose a categorization of the ML/DL approaches in medicine
- We comprehensively classify the most relevant medicine-related ML/DL applications

Let us sketch the principal assumptions in our work (and, consequently, the limitations of this survey): (i) we focus only on aspects concerning computer science, thus, we do not consider contributions from the Italian community focused on medical, ethical, and legal aspects; (ii) we focus principally on papers published in journals; and (iii) our analysis is based only on the SCOPUS databases. A more detailed discussion about such assumptions can be found in Section 2.2.

The paper is organized as follows: in Section 2, we present and discuss the methods used to gather the data for building this review. In particular, Section 2.1 describes the framework for the paper selection, Section 2.2 analyzes the possible limitations of our work, Section 2.3 presents the dimension used for the paper classification, and finally, in Section 2.4, we provide a description of the background. In Section 3, we present the output of our analyses. In Section 3.1, we present a general picture of the ML/DL research in Italy since 2018. Then, in Section 3.2, a comprehensive overview and classification of the ML/DL relevant papers in medicine are provided. Finally, in Section 4 we present our final considerations.

## 2. Methods

In this section, we describe our methods: first, we illustrate the ground of our survey (Section 2.1) and discuss the limitations (Section 2.2), and then, we present the dimension of our analysis (Section 2.3). Before starting with the description of the proposed framework, it is worth noting that the adoption of ML and DL methodologies to other research fields, apart from the medical and health domains, goes far beyond the results that are provided in this survey. To give an idea of the order of magnitude of the phenomenon, it it possible to observe that at the time of writing, more than 680,000 documents that include the words "machine" and "learning" or "deep" and "learning" have been indexed by SCOPUS after 2017, and more than 22,000 of them are represented by review and survey articles, with more than 2000 citations for the highest-cited ones (https://www.shorturl.at/lDPRT (accessed on 1 August 2023)). Among such reviews, it is possible to find applications of ML and DL methodologies to a set of very heterogeneous domains including, to cite some, environment and energy applications [3–5], materials science [6,7], agriculture [8,9], natural language processing and computer vision [10,11], industrial applications and IoT [12–15], and public health policies [16,17].

### 2.1. The Framework

We outline here the process for selecting and analyzing the papers included in the review. In Figure 3, we provide a visual representation of our methodology. The blue ovals represent the activities that were performed automatically, the green boxes are activities that were performed manually, and in the orange boxes, the number of papers outputted by the previous activity is reported.

The starting point of our work is the output of the query in Figure 3, performed (All the queries were performed on 13 January 2023) via SCOPUS (i.e., https://www.scopus.com/search/form.uri?display=advanced (accessed on 13 January 2023)), shown below:

```
(( TITLE-ABS-KEY (machine  AND  learning)
OR  TITLE-ABS-KEY (deep  AND  learning))
AND  (TITLE-ABS-KEY (medicine)
OR  TITLE-ABS-KEY (medical)  OR  TITLE-ABS-KEY (health)))
AND  PUBYEAR  >  2017
AND LIMIT-TO ( AFFILCOUNTRY ,  "Italy" )
```

We have selected all papers that in the title, in the abstract, or in the keywords present the term "machine learning" or the term "deep learning" and a reference to a medical area (i.e., one of the terms "medicine", "medical", and "health") and that are published

from 2018 (i.e., PUBYEAR > 2017) and at least one of authors has an affiliation to Italy (i.e., LIMIT-TO ( AFFILCOUNTRY , "Italy" )) (Notably, the data for Figures 1 and 2 are produced with similar queries, where the constraints about the publishing year and the affiliation country are properly removed).

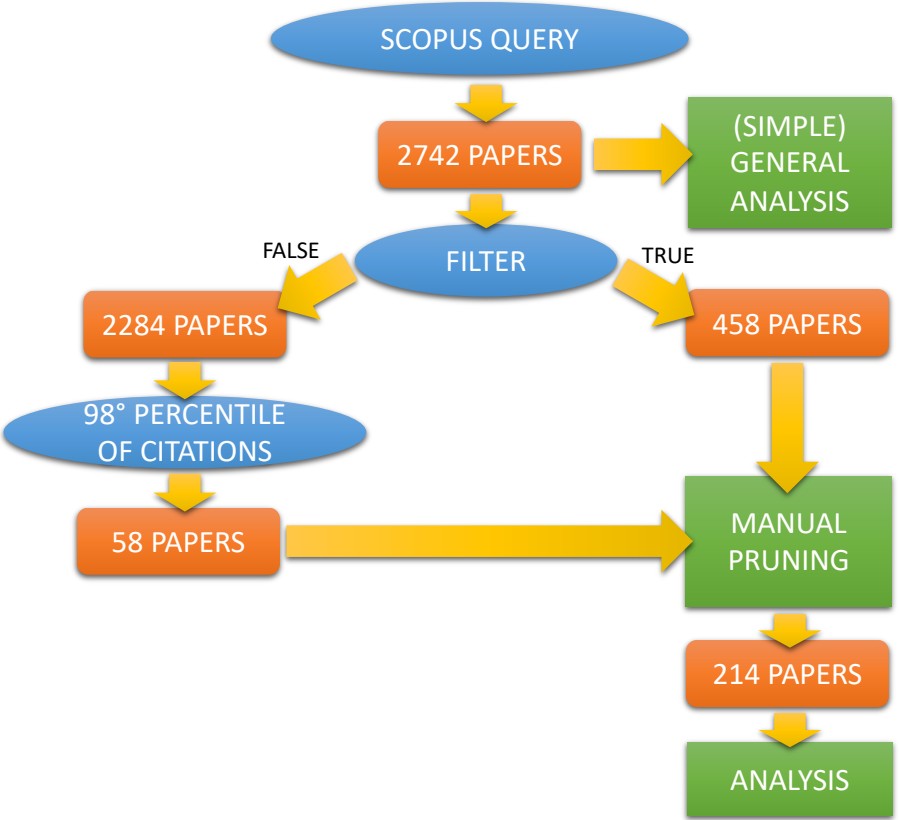

**Figure 3.** Graphical view of the framework applied in this work.

This query returns as output 2742 papers. These papers are used to provide a general description of the Italian research area in the field (see Section 3.1). Then, we filter (i.e., apply more conditions to the SCOPUS query) the papers to focus our attention on the works that are more significant and relevant to our analysis. We have restricted the papers to be analyzed on the basis of:

- Type of paper: only research journal papers (i.e., we excluded review/survey papers and conference papers);
- Subject area: we have considered only the relevant subject areas in SCOPUS, i.e., medicine, computer science, engineering, biochemistry, genetics and molecular biology, neuroscience, pharmacology, toxicology and pharmaceutics, health profession, nursing, dentistry, immunology and microbiology, and multidisciplinary.

Moreover, we considered only journals that have published at least 5 five eligible papers in the period (i.e., the papers are published in 38 journals).

Thus, we obtained a set of 458 papers that are considered for our analysis. These papers required manual pruning before we could begin the analysis phase. Some of these papers were not directly related to the medical field, (e.g., medical problems are only cited as possible applications of the proposed approaches, the query did not exclude all the reviews/surveys, there are position papers/letters about future perspectives, etc.). Additionally, we excluded any papers with international scientific collaborations in which it was clear that the Italian researchers did not contribute to the machine learning/deep learning aspects of the work. More specifically:

- If the paper includes the author contributions section, the contributions of the Italian researchers are checked to see if their work refers to topics regarding the development of ML/DL methodology, software, implementation, and so on. If such a case does not hold (e.g., the contribution from the Italian researcher is to provide data), the paper is excluded;
- If the paper does not include the author contributions section, for the Italian researchers, we checked both the department/institution they belong and the academic fields and disciplines on which they work (also by checking their track record and CV when available, and past articles they co-authored). If both of them are far outside the scientific fields in which ML and DL methodologies are usually developed and used, such as computer science, engineering, mathematics, statistics, and so on, the paper is excluded.

Next, we considered also 58 papers with a high number of citations (i.e., those in the 98th percentile) that were not selected by the previous filter. These papers were read and (manually) filtered in the same way.

After these filtering steps, we were left with 214 papers. We analyzed these papers using the criteria described in Section 2.3.

### 2.2. Limitations

Our survey may be, of course, subject to some limitations related to the research criteria we adopted. First, we focused our research on SCOPUS, thus, excluding a priori papers that were indexed by other databases such as Web Of Science (WOS) and PubMed only. However, we can safely assert that SCOPUS usually covers more journals and records than Web of Science and PubMed and that it usually represents the ideal source for this kind of research, also considering the huge coverage overlap among these databases. Furthermore, with regards to PubMed, it must be said that it is usually more oriented to the medical domain rather than to the computer science topics described in this survey.

Another limitation may be related to the research criteria adopted for selecting the relevant papers. In particular, we focused on the research articles published in international journals, thus, excluding conference proceedings, letters, reviews, and so on. The reasons for our choice were twofold. First, we wanted to concentrate on novel research only, and for this reason, reviews and surveys were excluded. Second, we excluded conference papers because, in our opinion, novel but well-established research is commonly published in scientific journals rather than presented at international conferences, especially with regards to the Italian research community. While this can be, in general, considered correct, in some cases, the outstanding research may also be presented at leading international conferences and/or included in articles of different types. To mitigate such issues, we also included seminal papers (i.e., 98 percentile top-cited articles) that were excluded by the filtering criteria described in the previous section. Such papers have a number of citations that ranges from 90 to more than 500.

Finally, it is worth noting that at the time of writing, the researchers belonging to the Italian community have contributed in the last five years to more than 40 books regarding the topics analyzed by this survey.

All of these facts suggest that the research community is fervent (see also the discussion in Section 3.1 about international funding and international co-authoring), even beyond the results shown by this survey.

### 2.3. Analysis Criteria

We analyzed the papers considering the following dimensions:

- the medical topics;
- the type of data;
- the type of pre-processing methods;
- the learning methods;
- the evaluation methods.

We also made some (simple quantitative) considerations about the publication journals.

To identify the most common medical topics addressed in the papers, we systematically recorded and analyzed them.

To classify the type of data used, we adapted the taxonomy proposed in [18] and included four categories: clinical images, biosignals, biomedicine, and electronic health records (EHR). However, we also encountered several types of data that did not fit within these four categories, which were sourced from diverse and problem-specific contexts. Given that the proportion of papers using these data was relatively low (around 10%) but relevant in the general context, we created a new generic class called "Others". Figure 4 provides a graphical representation of the taxonomy and its related sub-areas.

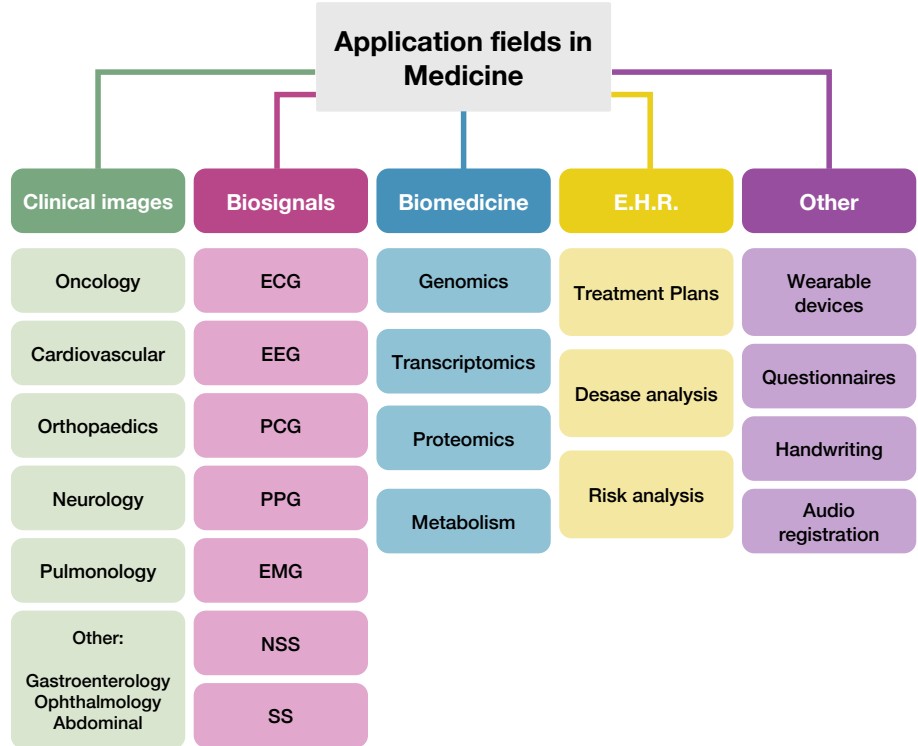

**Figure 4.** Graphical representation of the data type taxonomy in medicine for ML/DL.

Regarding the type of pre-processing methods and the learning methods, we classify the papers in the macro-areas according to the methods used. Note here that papers may belong to multiple classes if they encompass the use of methodologies belonging to different macro-areas.

We consider the following macro-classes for the type of pre-processing methods:

- feature selection;
- feature extraction;
- feature reduction;
- data filtering;
- data normalization;
- missing data management;
- undersampling;
- oversampling;
- other.

We instead consider the following macro-class for the type of learning methods:

- ML supervised;
- ML unsupervised;
- ML semisupervised;

- ML reinforcement learning;
- DL supervised;
- DL unsupervised;
- DL semisupervised;
- DL reinforcement learning.

Finally, we analyze the evaluation methods used to check the quality of the ML/DL models in the different papers. We aim to find possible gold standards and/or best practices in ML/DL model(s).

## 2.4. Background

In this section, we describe the basic concepts that are useful to have full comprehension for all readers (in particular for readers who are not familiar with the area of ML and DL).

First, we provide a brief description of the type of pre-processing methods:

- feature selection: is the process of reducing the number of input features (i.e., selection of the relevant features);
- feature extraction: is the process of manipulating and transforming (a subset of) the raw data into (new) features;
- feature reduction: is the process of data transformation from a high-dimensional space into a low-dimensional space, without losing their relevant properties in the transformation;
- data filtering: is the process of removing information that is not useful in a (large) database on a specific criterion;
- data normalization: is the process of transforming data into a standard format;
- missing data management: is the process of managing features that present missing values;
- undersampling and oversampling: is the process of adjusting (i.e., resampling) the class distribution of a data set. In the case of undersampling, the distribution of (observations/instances of) a class is artificially decreased. In the case of oversampling, the distribution of (observations/instances of) a class is artificially increased.

Notably, pre-processing is an optional activity that aims to manage the raw data and improve their quality with the goal of improving the performance of the learning process.

Then, let us point out that "machine learning" (ML) refers to classic machine learning such as, e.g., decision tree, regression, Bayesian classification (see for a detailed description, e.g., [19]) On the other hand, "deep learning" (DL) refers to an approach based on neural networks where the term "deep" points out the fact that the data are transformed through numerous layers (see for a detailed description, e.g., [20]).

Let us illustrate the main difference between classical ML and DL (for a detailed discussion, see [21]):

- DL needs a large amount of data, while ML can also work with a small amount of data;
- in DL, the feature engineering phase is eliminated; DL can learn features for improving the output accuracy. On the other side, in ML, the feature engineering phase is application dependent and manually performed;
- DL usually has higher computational requirements that often entail the adoption of specialized hardware with very high performance (i.e., DL has higher time requirements for training than ML);
- DL has mainly black box approaches (explainability is very difficult because of hyperparameters and complex network design); on the other side, ML provides also white box approaches;
- DL can achieve an accuracy rate higher than ML.

For ML and DL approaches, We consider four sub-classes (notably these sub-classes are orthogonal to the distinction between ML and DL):

- supervised: is an approach where the learning algorithm is trained on input data labeled with the correct output classes;
- unsupervised: is an approach where the learning algorithm is trained exclusively on unlabeled data. This is particularly useful when the outcome class is not known or the labeling process is too long and expansive;
- semisupervised: is a paradigm where the learning algorithm is trained by using a small amount of labeled data followed by a large amount of unlabeled data. This is particularly useful when a huge amount of data are available, but only a small fraction of them have been labeled;
- reinforcement learning: is a paradigm where the learning algorithm is self-trained on reward and punishment mechanisms (i.e., take actions and learn through trial and error).

A brief review of the methodologies and the types of ML and DL can be found in [21,22], while further details are available from various textbooks, such as the one by C. Bishop et al. [23] or the one by I. Goodfellow et al. [24].

## 3. Results

In this section, we present the output of our analyses:

- in Section 3.1, we provide a general analysis on all papers to provide a (simple) general snapshot of the ML/DL Italian research in the medical area;
- in Section 3.2, we provide a systematic analysis of the selected papers, as described in Section 2.

### 3.1. A Description of Italian Machine Learning/Deep Learning Research in the Medical Area at a Glance

In this section, we give a general description of the whole Italian research community through the papers published and indexed by SCOPUS since 2018.

As described in Section 1, the Italian community is one of the most productive players in the area with more than 2500 papers. Figure 5 shows a continuous increasing trend in the last 5 years. It is quite interesting to point out that most papers are published in international journals (i.e., approximately 74%) and are open-access (i.e., approximately 59%). In these papers, 74 Italian institutions are involved: in this list, there are not only universities and research institutes but also hospitals. This fact shows that participation is wide-ranging and concerns actors in all aspects, i.e., both AI and medical ones, and shows a link between the research and academic groups with the local communities.

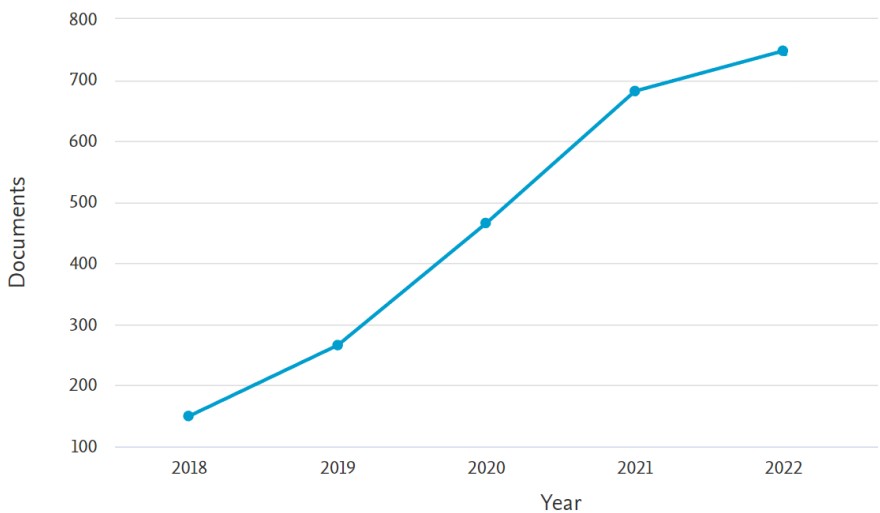

**Figure 5.** Report of documents published by the Italian community in machine learning/deep learning for the medical field in the years 2018–2022, indexed by SCOPUS.

Analyzing the funding sponsor's point of view, it is possible to see a wide spectrum of national and international funding sponsors. Figure 6 shows the top 10 funding sponsors whom the papers have acknowledged. From a numerical point of view, the European Union was the first sponsor. Notably, the voice "European Union" groups different types of grants, e.g., Horizon 2020, the 7th Framework Programme for Research, and the European Research Council. Moreover, the financial sponsorship of the Italian government is very relevant through grants provided by two different ministries (i.e., Ministry of Education, University and Research, and Ministry of Health). The other funding sponsors confirm well-established participation in international projects funded by grants, particularly from U.S.A. and U.K. agencies. Notably, the Italian research community's international involvement is very high and this fact is confirmed by the data concerning the nationality affiliation of the co-authors, see Figure 7.

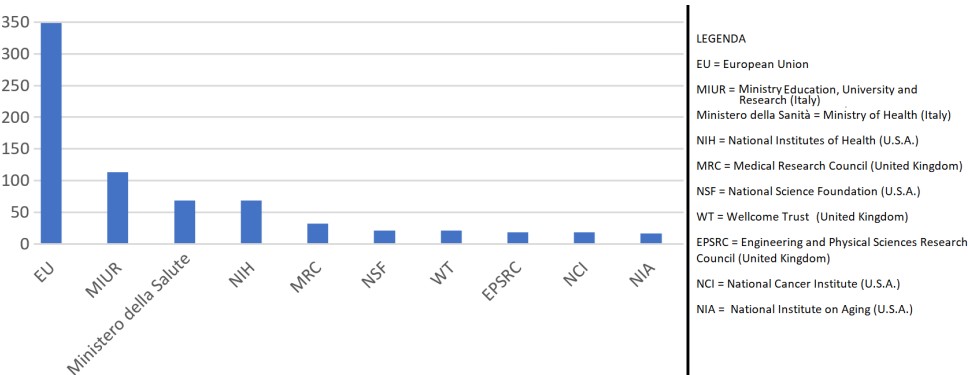

**Figure 6.** The top 10 funding sponsors acknowledged in the papers.

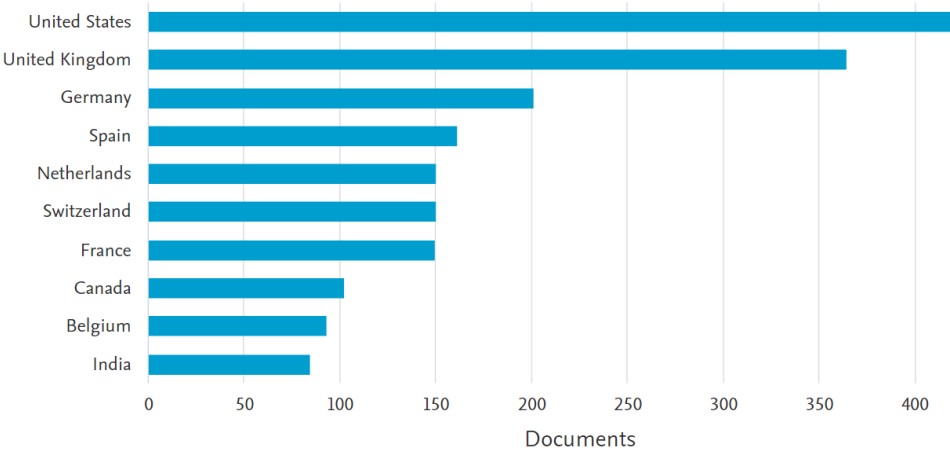

**Figure 7.** The top-10 coauthoring countries with Italian researchers.

### 3.2. Systematic Analysis

On the basis of the criteria and the "manual pruning" phase described in Section 2.1, we have selected 224 papers. These papers were analyzed through the criteria described in Section 2.3.

Figure 8 shows the source journals of 224 papers. The papers are distributed in 42 journals. In Figure 8, the journals are ordered in an alphabetic way. In the period analyzed, the top 3 journals for the number of publications are IEEE Access, Scientific Reports, and Applied Science.

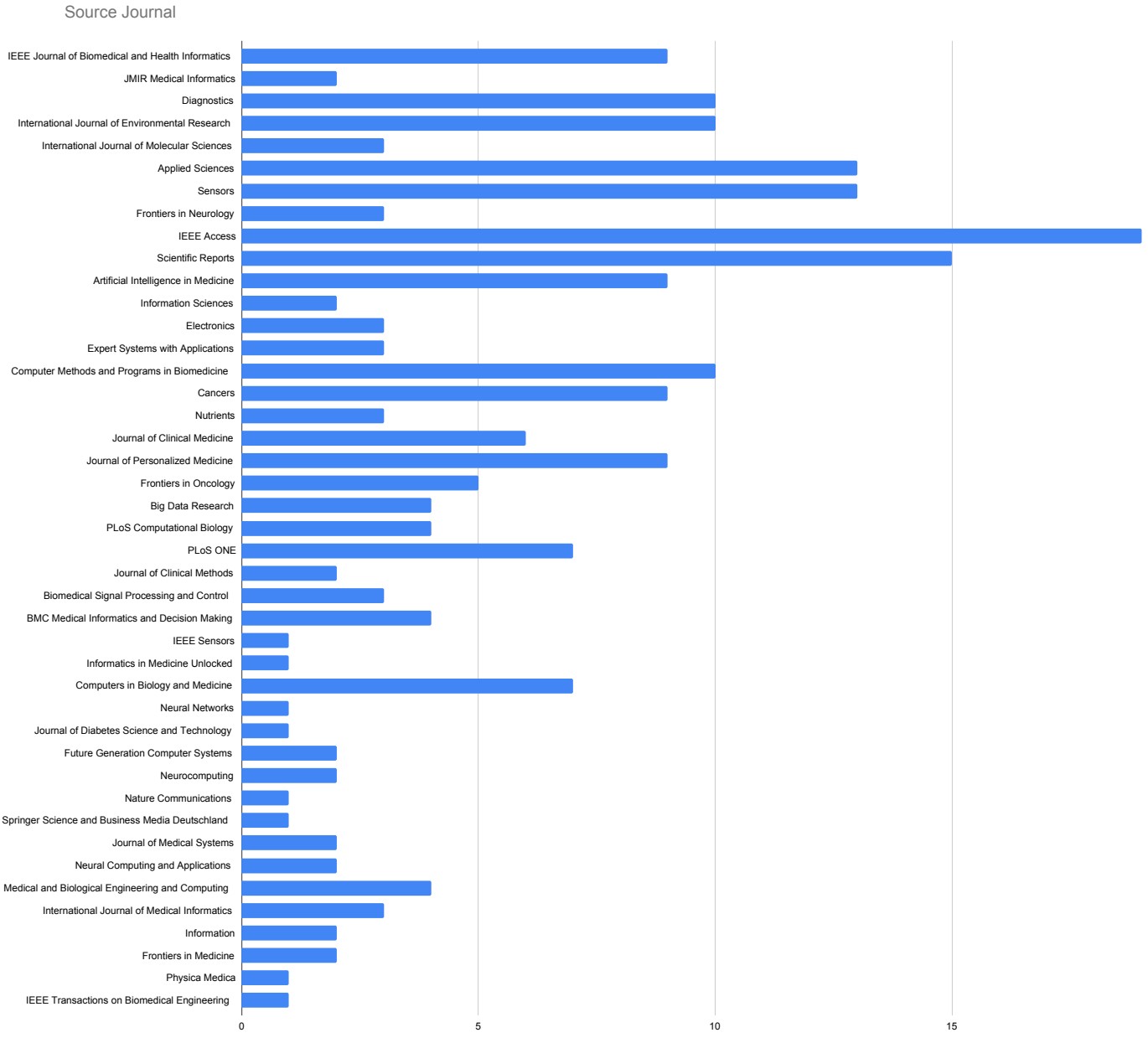

**Figure 8.** Source journals of papers considered in the systematic review.

Notably, not all the 224 papers are analyzed with our methodology, since 10 of these (i.e., [25–34]) proposed new ML/DL methodologies, metrics, or approaches that may be strongly related to the medical area, but they are not focused on a specific disease or on a particular case study. However, we believe that it is very important that the Italian community not only provides a bridge between the ML/DL area and the medical area but also proposes solutions to the general issues, which arise from the peculiarity of the medical field.

First, we focused on the medical topics considered in the paper we have analyzed. Figure 9 shows the distribution of these medical topics and Table 1 reports the paper classification by medical topic. Note that all the topics considered only by just one paper are included within the *other* category.

**Table 1.** Classification by medical topic.

| Topic | Reference |
| --- | --- |
| Alzheimer's disease | [35–40] |
| Autism Spectrum Disorders | [41,42] |
| Brain Tumors | [43–45] |
| Breast Cancer | [46–60] |
| Cardiovascular disease | [61–65] |
| Chronic Kidney Disease | [66–68] |
| Dementia | [69–72] |
| Diabetes | [73–78] |
| Exposure to extremely low frequency waves | [79,80] |
| Glioblastoma | [81,82] |
| Heart Failure | [83–87] |
| Kidney Disease | [88,89] |
| Lung Cancer | [90–94] |
| Melanoma | [95,96] |
| Multiple Sclerosis | [97–99] |
| Parkinson's Disease | [100–107] |
| Prostate Cancer | [108,109] |
| Rectal Cancer | [110–112] |
| SARS-CoV-2 | [113–136] |
| Seasonal Flu | [137,138] |
| Sepsis | [139–141] |
| Stroke | [142,143] |
| Varicella Zoster | [144,145] |
| Voice-related Pathologies | [146,147] |
| Other Types of Cancer | [148–156] |
| Surgery-Related | [157–164] |
| M-health | [165–170] |
| Patient Telemonitoring | [171–174] |
| Liver Diseases | [175–177] |
| Orthopedic | [178–180] |
| Arterial Disease | [181–183] |
| Trauma | [184,185] |
| Other | [186–247] |

The most-faced topic is represented by *SARS-CoV-2* (i.e., 11.2%). This result is not strange, since we considered the pandemic period, and most of the efforts of the scientific community were focused against SARS-CoV-2. *Cancer* is also a very important topic (i.e., 16.77%), with *breast cancer* representing one of the most considered topics (i.e., 7%), along with lung cancer, prostate cancer, and colorectal cancer, which are considered by more than one paper. Such results are in line with cancer incidence, which sees these cancers as the most common types of cancers for occurrence in Europe [248]. Since different types of cancers are considered, we considered those appearing in just one article in the

*other types of cancer* category (i.e., 4.2%). Another relevant topic is represented by *Parkinson's disease* (i.e., 3.7%).

**Figure 9.** Medical topics considered in the papers. Percentages are rounded to the first decimal place.

Then, we analyzed the dimension of the *type of data* used. Table 2 shows that the majority of the papers (i.e., 82.2%, 176 papers) use only one data type. However, it is quite interesting to note that 38 papers (i.e., 17.5%) use 2 or 3 different types of data, dealing with the issue of managing data with different characteristics.

Figure 10 shows the distribution of the type of data used in the paper we analyzed. The most used type of data is E.H.R., and the second one is clinical images. We point out that the class *other* reaches the value of 16.3%, which is higher than the *Biomedicine* and *Biosignals* categories. This fact indicates that several medical topics and diseases involve the need to manage a lot of different types of data to assess and characterize their complex features.

**Table 2.** Number of data types used.

| Number of Data Types | Number of Papers |
| --- | --- |
| 1 | 176 |
| 2 | 32 |
| 3 | 6 |

With regards to the *pre-processing* category, Table 3 shows that the majority of the papers (i.e., 128, approximately 60%) indicate that the authors applied at least one pre-processing technique.

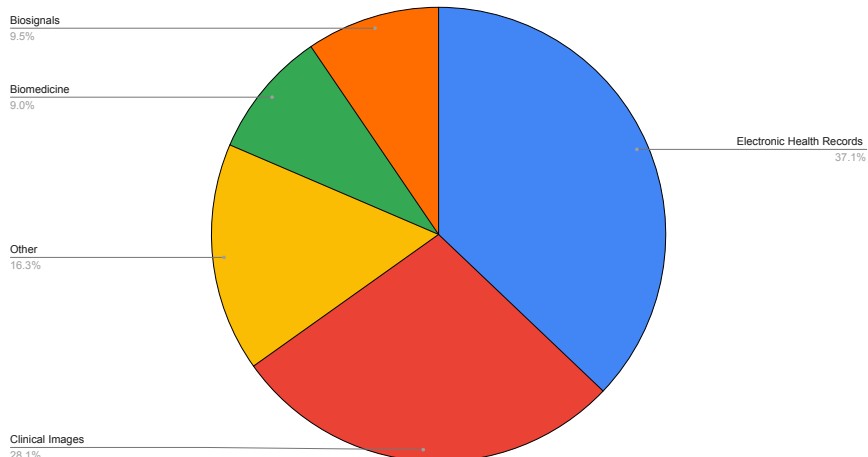

**Figure 10.** Distribution of data types used. Percentages are rounded to the first decimal place.

**Table 3.** Number of pre-processing methods used.

| Number of Pre-Processing Methods | Number of Papers |
| --- | --- |
| 0 | 86 |
| 1 | 87 |
| 2 | 29 |
| 3 | 10 |
| 4 | 2 |

Figure 11 shows the distribution of the principal pre-processing techniques. We can see that *Feature selection* and *Feature extraction* are the most used techniques, covering approximately 50% of all cases.

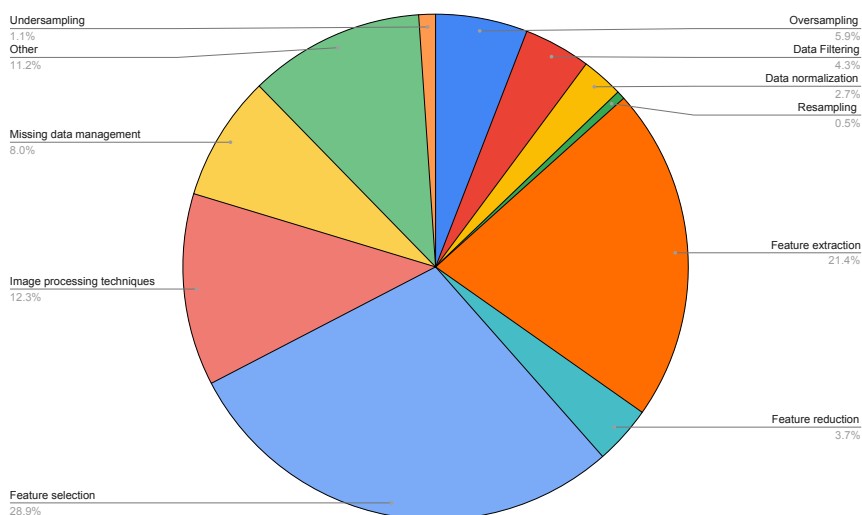

**Figure 11.** Distribution of pre-processing types used. Percentages are rounded to the first decimal place.

Figure 12 shows the distribution of ML/DL methodologies adopted, and Table 4 presents the papers' categorization by methodology. As described in Section 2.3, we cataloged the approaches used in 8 macro-categories.

**Table 4.** Classification by approach.

| Methodology | Approach | Reference |
|---|---|---|
| Machine Learning | Unsupervised | [47,51,54,58,60,79,80,94,98,186,187] [107,118–120,138,149,155,173,174,188,189] |
| | Supervised | [35,38,39,181,182,190–194] [42,44,48,53,55,56,125,165,195,196] [57,59,61,64,65,149–151,157,197] [67,68,152,153,166,183,184,198–200] [69,70,72,74–77,89,167,201] [78,81,83–85,175,178,202–204] [88,90,91,158,176,177,205–208] [92,93,95,96,99,169,172,209–211] [100,101,160,179,212–217] [102–106,108–111,161,173,180,218–220] [113,115–117,120,123,125–128,221–224] [130,132–134,136,137,139–141,156] [143,144,162,163,170,185,225–229] [40,49,62,71,146,147,168,186,230,231] [87,112,122,124,129,228,232,233] |
| | Semi-Supervised | [40,43,119,234] |
| | Reinforcement Learning | [37,129,235] |
| Deep Learning | Unsupervised | [49,125,135,159,189,197] |
| | Supervised | [45,46,50,52,62,63,66,148,236,237] [71,73,82,86,168,230–232,238,239] [87,97,112,154,171,233,240–243] [114,121,122,124,131,142,145,164,244–247] [39,44,77,190,191,208,209,213,234] [101,115,132,135,141,161,214,218,222,224] |
| | Semi-Supervised | [174] |
| | Reinforcement Learning | [94,148] |

Research that adopts multiple approaches may be present in more than one line.

The majority of the papers (i.e., 171, approximately 80%) use one or more approaches belonging to only one of these categories, whereas the remaining papers (i.e., 43, about 20%) use two or more approaches belonging to two categories. It is quite interesting to note that the ML approaches (i.e., 72.8%) are more used than DL approaches, and the most used approach belongs to the ML supervised category (i.e., 62.5%). In general, supervised approaches, including both ML and DL, are largely adopted (i.e., 86.3%). These facts underline how ML approaches still represent the most used approaches in the medical field in Italy, probably due to the scarcity of data needed to train DL approaches, which are, however, widely applied for image-related problems. Furthermore, most learning methodologies are supervised and, thus, focused towards a specific outcome that is already present in the training data and that clearly determines the medical question that the model should address.

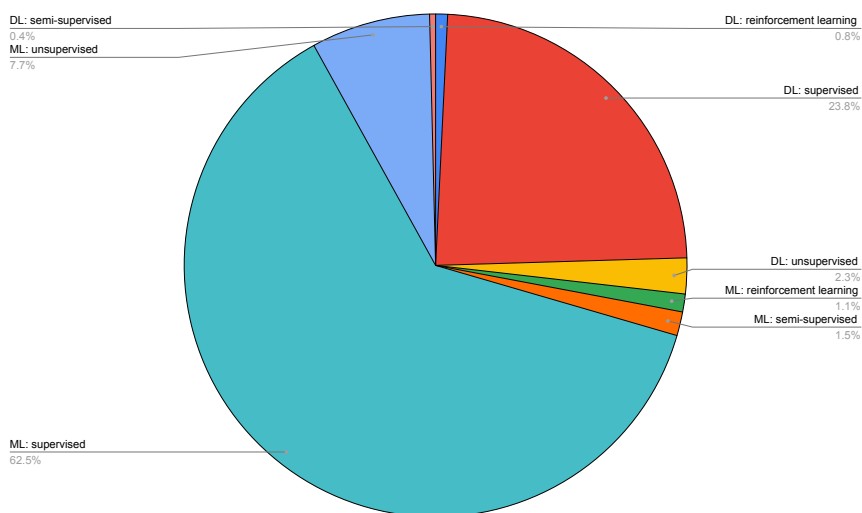

**Figure 12.** Distribution of the used ML/DL approaches. Percentages are rounded to the first decimal place.

Finally, we also analyzed the evaluation methodology employed in the papers to check the quality of the model(s) built. The analysis performed shows that there is not a de facto gold standard for this activity within the research community for this aspect. As a matter of fact, there is a significant heterogeneity among the techniques and statistical measures used by the different authors. However, we can summarize our analysis of this aspect on two topics:

- how the validation phase is performed;
- the statistical measures used.

Regarding the validation phase, we observed that such a phase is predominantly (though not exclusively) performed using two main methods:

- a fixed split of the dataset into a training set and a test set, as well as k-fold cross-validation. The fixed split method was employed in 49 papers, accounting for approximately 22.9%. The most commonly used split values were 90–10%, 80–20%, and 70–30%;
- the k-fold cross-validation method was used in 99 papers, representing approximately 46% of the sample. Various values of k were utilized, but the most frequently used were 10, 5, and 3.

Regarding the statistical measures employed, we observed a wide variety of measures and a lot of different combinations of these measures. However, we observed that three measures stood out as the most commonly used but not globally used: accuracy, ROC-AUC, and F1-score. Accuracy was used in approximately 112 papers (approximately 52.3%), ROC-AUC in 81 papers (approximately 37.6%), and F1-score in 66 papers (approximately 31%). Once again, the absence of a gold standard in the selection of statistical measures for evaluating trained ML/DL models becomes apparent.

In conclusion, our analysis highlights the lack of consensus in the research community regarding the choice of evaluation techniques and statistical measures for assessing ML/DL models. However, this finding highlights the prevalence of specific evaluation methods that could be considered potential best practices within the research community.

## 4. Discussion

The analysis of the state of the art in the scientific papers focusing on ML and DL for medicine over the last five years has uncovered a rapidly expanding research area with substantial potential for applications in healthcare (see Figure 1).

First, we proposed a methodological analysis for the papers indexed in SCOPUS, identifying a common set of dimensions. Our analysis encompassed a total of 2742 papers, out of which we conducted a detailed methodological examination of 516 papers. Among these, 214 were studied using the dimension we proposed. The findings (see Section 3) provided a comprehensive overview of the Italian research landscape in this field. In particular, these findings show how the Italian research community has worked on a very heterogeneous range of medical problems (see Table 1).

As pointed out in Figure 2, Italy had a very important role with regards to the use and application of ML and DL methodologies to the medical field in the last five years, being among the first 10 countries for the number of documents indexed by SCOPUS. Moreover, if we take a look at the gross domestic spending on the R&D in these countries, according to the available data provided by the Organisation for Economic Co-operation and Development (OECD) (source: https://data.oecd.org/rd/gross-domestic-spending-on-r-d.htm (accessed on 24 August 2023)), we see that Italy typically invests in R&D less than the other considered countries, with only the exception of Australia. If we limit our comparison to the European community landscape, by comparing the latest available data, Italian spending on R&D is approximately one-quarter of the German spending and approximately one-half of the French one (129,005.3 Million US dollars in 2021 for Germany, 63,542.7 in 2021 for France and 33,805.9 in 2021 for Italy). Furthermore, UK spending seems to be much higher (78,153.30 Million US dollars in 2020). This situation does not change much if we limit our research to government researchers in France and Italy that show very similar values in 2018, but Germany invests more than the sum of the two (always according to the available data). This may underline the great effort, at least in quantitative terms, of the Italian researchers in the field of ML and DL applied to the medical domain. Clearly, this rather impromptu comparison does not tell us all the truth, and also, qualitative measures should be taken into account for a fair and complete comparison. However, this goes far beyond the scope of this survey.

It is noteworthy to emphasize that a survey on the use of ML and DL in the medical field in Italy holds the potential to appeal to a wide range of readers (i.e., stakeholders) owing to a multitude of distinct rationales. In particular, these potential stakeholders could include:

- Medical and Healthcare Professionals: they could be interested in understanding how ML and DL are used in practice and have reference to specific medical topics that are studied in the research;
- Healthcare institutions: they could find indications about the treatment of specific disease(s) in their interest and consider the adoption of these technologies;
- Technology Companies: when developing AI-based solutions, they may be interested in understanding the needs of the Italian medical sector and in finding possible collaborations in the Italian academic community;
- Patient Associations: they could know how ML and DL are used to improve the quality of care and assistance they receive;
- Academics institutions and Researchers: they could be interested in a detailed review of the current applications of using ML and ML in medical practice to contribute to future research and innovation and to find possible partners for new research projects;

Finally, it is really important to acknowledge that the use of ML and DL methodologies raises several legal and ethical concerns. The analysis and discussion on these topics are out of the scope of this paper. However, let us point out that the growing interest in and the adoption of ML/DL systems in the medical field, along with the positive results obtained, indicate the potential for these systems to serve as valuable tools in laboratory settings in the coming years.

**Author Contributions:** Conceptualization, A.B. and M.P.; methodology, A.B. and M.P.; investigation, A.B. and M.P.; data curation, A.B. and M.P.; writing—original draft preparation, A.B. and M.P.; writing—review and editing, A.B. and M.P.; visualization, A.B. and M.P.; supervision, A.B. and M.P. All authors have read and agreed to the published version of the manuscript.

**Funding:** This research received no external funding.

**Conflicts of Interest:** The authors declare no conflict of interest.

## Abbreviations

The following abbreviations are used in this manuscript:

| | |
|---|---|
| AI | Artificial intelligence |
| AUC-ROC | Area under the curve receiver operating characteristic |
| DL | Deep learning |
| IoT | Internet of Things |
| ML | Machine learning |

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
