# Peer review of "Exploring the State of Machine Learning and Deep Learning in Medicine: A Survey of the Italian Research Community"

_information, doi:10.3390/info14090513_

Round 1
Reviewer 1 Report
In this manuscript the authors discuss about the evolution of ML and DL in the context of Medicine. They refer specifically to the Italian community on top of the AI evolution, regarding research works that took place during the last 5 years.
Overall, a good article that tackles an interesting topic. It has however some points that require additional content, as follows.
1. In the abstract please refer to the challenges and advancements offered through AI, ML, and DL.
2. Moreover, in the abstract, add some additional details regarding the conducted study – what exactly did you research for the latest five years? What were your quality criteria during this research? Did you have any assumptions?
3. In the introduction, I would like to see more information regarding the pains and gains of AI
4. Why big data and IoT revolutionized the studied field? This is not clearly identified
5. What made the authors to describe the evolution of AI specifically in Italy and not other country out of the other top 7 countries in this domain?
6. In the introduction, could you please refer to the assumptions that were made during your study?
7. Before explaining your framework in the Methods section, I would like a brief introduction to the advancements of AI, ML, and DL in other domains – not only healthcare and medicine. For this you could cite and refer to the following articles and research projects:
a. Biran, Ofer, et al. "PolicyCLOUD: A prototype of a cloud serverless ecosystem for policy analytics." Data & Policy 4 (2022): e44.
b. Kyriazis, Dimosthenis, et al. "The CrowdHEALTH project and the hollistic health records: Collective wisdom driving public health policies." Acta Informatica Medica 27.5 (2019): 369.
c. Beattie, Alexander, et al. "A Robust and Explainable Data-Driven Anomaly Detection Approach For Power Electronics." 2022 IEEE International Conference on Communications, Control, and Computing Technologies for Smart Grids (SmartGridComm). IEEE, 2022.
d. Manias, George, et al. "SemAI: A novel approach for achieving enhanced semantic interoperability in public policies." Artificial Intelligence Applications and Innovations: 17th IFIP WG 12.5 International Conference, AIAI 2021, Hersonissos, Crete, Greece, June 25–27, 2021, Proceedings 17. Springer International Publishing, 2021.
e. Bernault, Chloé, et al. "Assessing the impact of cognitive biases in AI project development." International Conference on Human-Computer Interaction. Cham: Springer Nature Switzerland, 2023.
8. Why did you use only the keywords of ML and DL during your study? Why did not you also use the word of AI?
9. Figure 8 is not visible – please improve it
10. Regarding the discussion of the results, I am missing a clear statement regarding your final outcomes
11. Regarding Section 4, please provide your next steps and try to include the potential stakeholders/receivers of this work.
12. How are you going to communicate and disseminate your research findings? This should be also listed in Section 4
Reviewer 2 Report
The paper provides a comprehensive survey of the Italian research community's use of Machine Learning (ML) and Deep Learning (DL) techniques in the medical field. The paper is organized into 4 sections, including a description of the methods used to gather data for the review, a general analysis of ML/DL Italian research in the medical area, and a systematic analysis of selected papers.
The authors note that the medical field has always been a source of challenges and an important area for both experimenting and developing AI methodologies. Even one of the first and most prominent AI research areas is Machine Learning (ML) in the past century, the development and use of ML methodologies in medicine were limited. However, with the abundance of health-related data, combined with advances in computing power and algorithms, the use of AI in medicine has grown significantly. The survey highlights the growing importance of Artificial Intelligence (AI) in the medical field and the increasing adoption of ML and DL techniques.
The authors provide examples of specific applications of ML and DL in medicine, including image analysis, diagnosis, and treatment planning.
The paper provides a detailed analysis of the selected papers, including the type of data used, the ML/DL techniques employed, and the specific medical applications.
The authors emphasize the importance of collaboration between medical professionals and AI experts to ensure that AI is used in a responsible and effective manner.
However, before publication few issues should be addressed carefully:
- Regarding the methodology for selecting the papers, if your query returns a paper with hundreds of authors and one of them is Italian do you consider it part of Italian research? The methodology should be more clearly presented emphasizing the criteria for inclusion/exclusion of the papers and their justification.
- How do the results presented in this paper compare with other similar studies? Is the Italian research community more active in the field of ML than others? We have no term of comparison.
- You have 228 references which is bewildering. I understood that 224 are the papers found. Thus, there are only 4 references for the whole paper: methodology, data processing etc. I recommend not to include the papers found on SCOPUS as references, you can change table 2 and instead of the “Reference” column use “Number of papers found” without referring to them. Then you can add them in an annex with all the details. In the references section include only the paper related to the “research” part of your work. I can’t imagine a possible reader of your paper going through 228 references!
- Be more careful with the references list. I found the expression “cited by x” for several references. This is not relevant for a reference list since some items have “cited by 0” and may raise the question why you include them.
- Improve the conclusion section: how is thew Italian research community compared with others?
The English is good.
Reviewer 3 Report
This is a comprehensive and informative paper on applying ML/DL methodologies in medical research. Although the citations are restrictive to Italy, the analytical approach to AI-based publications has covered essential domains of scientific accomplishments made by Italian scientists. The paper could elaborate more details on the following areas:
1. Differences and Similarities in ML/DL methods should be noted.
2. The evaluation of diverse approaches to the goodness-of-fit models derived from ML/DL should be discussed.
3. Please clarify the difference between founding and funding sources. I assume that you are centered on "funding".
This is a fine study that could be carefully amended and edited to improve its readability.
It is readable.
Round 2
Reviewer 2 Report
I have reviewed the revised manuscript and I can say that the authors have addressed all of the concerns I raised in my previous review. The paper is now in a much stronger position and I believe it is ready for publication.
In particular, I appreciate the authors' efforts to clarify the research questions, strengthen the methodology, and improve the presentation of the results. I believe these changes have made the paper more accessible and persuasive.
Therefore my recommendation is to accept the paper for publication.
The English of the paper is good enough.